# The SRG rat, a Sprague-Dawley Rag2/Il2rg double-knockout validated for human tumor oncology studies

**Fallon K. Noto[1]☺\*, Jaya Sangodkar[2]☺, Bisoye Towobola Adedeji[1], Sam Moody[1], Christopher B. McClain[1], Ming Tong[3], Eric Ostertag[3], Jack Crawford[1], Xiaohua Gao[2], Lauren Hurst[2], Caitlin M. O'Connor[2], Erika N. Hanson[2], Sudeh Izadmehr[4], Rita Tohmé[5,6], Jyothsna Narla[7], Kristin LeSueur[8], Kajari Bhattacharya[8], Amit Rupani[8], Marwan K. Tayeh[8], Jeffrey W. Innis[8,9,10], Matthew D. Galsky[4], B. Mark Evers[11], Analisa DiFeo[12], Goutham Narla[1,2‡], Tseten Y. Jamling[1‡]**

**1** Hera BioLabs Inc., Lexington, Kentucky, United States of America, **2** Division of Genetic Medicine, Department of Medicine, The University of Michigan, Ann Arbor, Michigan, United States of America, **3** Poseida Therapeutics Inc., San Diego, California, United States of America, **4** Division of Hematology and Medical Oncology, Tisch Cancer Institute, Icahn School of Medicine at Mount Sinai, New York, New York, United States of America, **5** Case Comprehensive Cancer Center, Case Western Reserve University, Cleveland, Ohio, United States of America, **6** Case Western Reserve University, Cleveland, Ohio, United States of America, **7** Regional Medical Center, San Jose, California, United States of America, **8** Department of Pediatrics, The University of Michigan, Ann Arbor, Michigan, United States of America, **9** Department of Human Genetics, The University of Michigan, Ann Arbor, Michigan, United States of America, **10** Department of Internal Medicine, The University of Michigan, Ann Arbor, Michigan, United States of America, **11** Markey Cancer Center, University of Kentucky, Lexington, Kentucky, United States of America, **12** Department of Obstetrics and Gynecology, The University of Michigan, Ann Arbor, Michigan, United States of America

☺ These authors contributed equally to this work.
‡ GN and TYJ also contributed equally to this work.
\* fnoto@herabiolabs.com

**Data Availability Statement:** All relevant data are within the paper and its Supporting Information files.

## Abstract

We have created the immunodeficient SRG rat, a **S**prague-Dawley **R**ag2/Il2r**g** double knockout that lacks mature B cells, T cells, and circulating NK cells. This model has been tested and validated for use in oncology (SRG OncoRat®). The SRG rat demonstrates efficient tumor take rates and growth kinetics with different human cancer cell lines and PDXs. Although multiple immunodeficient rodent strains are available, some important human cancer cell lines exhibit poor tumor growth and high variability in those models. The VCaP prostate cancer model is one such cell line that engrafts unreliably and grows irregularly in existing models but displays over 90% engraftment rate in the SRG rat with uniform growth kinetics. Since rats can support much larger tumors than mice, the SRG rat is an attractive host for PDX establishment. Surgically resected NSCLC tissue from nine patients were implanted in SRG rats, seven of which engrafted and grew for an overall success rate of 78%. These developed into a large tumor volume, over 20,000 mm³ in the first passage, which would provide an ample source of tissue for characterization and/or subsequent passage into NSG mice for drug efficacy studies. Molecular characterization and histological analyses were performed for three PDX lines and showed high concordance between passages 1, 2 and 3 (P1, P2, P3), and the original patient sample. Our data suggest the SRG OncoRat is a valuable tool for establishing PDX banks and thus serves as an alternative to

**Funding:** NCI contract HHSN261201700017C
supported PDX establishment and their molecular
and histological characterizations, along with their
drug response studies. Hera Biolabs, Inc. provided
support in the form of salaries for authors (FKN,
BTA, SM, CM, JC, TYJ) and Poseida Therapeutics,
Inc. provided support in the form of salaries for
authors (EO, MT), but did not have any additional
role in the study design, data collection and
analysis, decision to publish, or preparation of the
manuscript. The specific roles of these authors are
articulated in the 'author contributions' section."
Support: Sequencing studies were carried out with
the support of the Michigan Medical Genetics
Laboratories in the Department of Pediatrics and
the support of C.S. Mott Children's Hospital.PDX
research was supported by the Biospecimen
Procurement & Translational Pathology Shared
Resource Facility of the University of Kentucky
Markey Cancer Center (P30CA177558).

**Competing interests:** Dr. Goutham Narla is on the
scientific advisory board for HERA Biolabs. This
does not alter our adherence to PLOS ONE policies
on sharing data and materials.

current PDX mouse models hindered by low engraftment rates, slow tumor growth kinetics, and multiple passages to develop adequate tissue banks.

## Introduction

*In vivo* models are essential in determining the effectiveness and safety of potential treatments prior to clinical testing in patients. These preclinical models provide critical information on the toxicity and efficacy of novel drugs and allow researchers to identify and address potential areas for further pharmacological and biological optimization. Moreover, patient-derived xenografts (PDX), in which tumor tissue is taken directly from the patient and grown in laboratory animals, may be more predictive than cell line-derived xenografts from established cell lines, as PDX models more closely recapitulate the histology and genomic features of the original tumor.

Immunodeficient mice have proven essential for the establishment of *in vivo* human tumor models. These mouse models demonstrate markedly variable differences in human cancer cell line uptake and growth kinetics which broadly determine the feasibility of conducting cancer therapeutic efficacy studies. One of the most robust and well characterized mouse models is the non-obese diabetic (NOD) Cg-*Prkdc*$^{scid}$ *IL2rg*$^{tm1Wjl}$/SzJ (NSG mouse). NSG mice are genetically engineered for knockout mutations in the Prkdc and Il2rg genes rendering the model B, T, and NK cell deficient [1–4]. In addition, the presence of NOD strain polymorphisms in the signal-regulatory protein alpha (Sirpa) gene of the NSG mouse enhances human immune system engraftment for immuno-oncology studies [5–7].

The ability to transfer and propagate human tumors in animal models allows for unique opportunities to study tumor biology, dissect the molecular mechanisms driving tumor development and progression, and evaluate treatment response. While patient derived xenograft (PDX) models are a valuable cancer model, there are some limitations in employing such model systems for precision medicine approaches. Low engraftment rates and slow growth for certain cancers like human non-small cell lung cancer (NSCLC) make it challenging to create these PDX models [8–10]. Even if successfully engrafted and established, multiple passages in the mouse are required to generate enough tissue for efficacy studies due to the small tumor volume supported by the mouse, which makes it technically very challenging to generate personalized drug-response data in a timely manner to affect patient treatment. Furthermore, studies have shown that genetic drift occurs across serial passages *in vivo* in transplanted PDXs as demonstrated by changes in the copy number alterations (CNA) landscape [11]. This in turn results in PDXs that no longer faithfully reflect the genomic landscape of the primary tumors.

Although mouse models have been instrumental for *in vivo* oncology testing, variable tumor uptake and differences in drug metabolism/physiology can hinder translation to humans. Mouse models therefore are not always ideal for drug efficacy testing and downstream analyses such as pharmacokinetics, pharmacodynamics, and toxicology. Since the rat is often the preferred rodent species for preclinical studies due to size and robust nature, particularly for pharmacokinetic and toxicology assessments, a severely immunodeficient rat model could be highly advantageous for oncology studies. Additionally, the rat could be an alternative rodent model for cell lines that present significant engraftment and growth challenges in the existing mouse models. Several strategies have been utilized to develop genomic alterations in rats [12–14]. Previously we reported a Rag2 (Recombination Activating Gene 2) knockout rat

on the Sprague-Dawley strain (SDR rat) which is mature B cell deficient and severely depleted of T cells [15]. SDR rats demonstrated high efficiency and desirable uniformity in a variety of human tumor growth profiles and grew tumors to nearly ten times the volume (or double the diameter) allowed in mice. Rats also accommodate serial blood and tumor tissue sampling for temporal assessment of several parameters from the same animal. For example, efficacy, pharmacokinetics, clinical pathology, toxicity endpoints, systemic exposure, and biomarker endpoints can all be collected from one animal at several timepoints.

Despite these advances, some important human cancer cell lines, such as the VCaP prostate model, exhibit high variability and poor tumor growth in both SDR rats and NSG mice, hindering the ability to run efficacy studies [16]. In order to overcome these deficiencies, we have created a rat with a functional deletion in both the Rag2 and Il2rg genes on the Sprague-Dawley background (SRG rat) that lacks B, T, and NK cells. The SRG rat supports the growth of multiple human cancer cell lines, including lines that do not engraft or grow well in existing mouse models, such as VCaP. In addition, SRG rats are highly permissive to engraftment with NSCLC-PDX tumors from patients. Here we highlight growth kinetics of several human cancer cell lines and NSCLC-PDX samples in the SRG rat.

Our data demonstrate that the SRG rat has the potential to be a valuable model for evaluating drug efficacy in a wide range of human cancers. Future uses for this model include developing a better understanding of the efficacy and toxicity of drug therapies and allowing for consistent and rapid translation from genomic findings to proof of concept *in vivo* studies. Our goal is the ultimate translation of these capabilities into the clinic.

## Methods

### FACS analysis of immune cells

To detect T, B, and NK cells in SRG rats, flow cytometric analysis was performed on splenocyte, thymocytes, and whole peripheral blood using a BD LSRII. Blood was collected in $K_2$EDTA tubes. Spleen and thymus were collected in FACS buffer (BD Pharmingen 554656). The tissues were homogenized and passed through a 70 μm cell strainer to remove clumps. Red blood cells in tissues and blood were lysed by incubating with ACK Lysing Buffer (Quality Biological #118-156-721) for 10 minutes at room temperature. Cells were stained with fluorophore-labeled antibodies at a final concentration of 25 μg/mL in 20 μL volume for 20 minutes. Antibodies used were PE mouse anti-rat IgM (BD Pharmingen #553888), APC Mouse anti-rat CD45R (Biolegend #202314), PE Mouse Anti-Rat CD8a (BD Pharmingen #559976), APC Mouse Anti-Rat CD4 (eBioscience #17-0040-80), and APC Mouse Anti-Rat CD161a (Biolegend #205606).

### Cell culture

Human VCaP (ATCC® CRL2876™), HCT-116 (ATCC® CCL247™), MIA-PaCa-2 (ATCC® CRL-1420™), HCC1954 (ATCC® CRL-2338™), and 786-O (ATCC® CRL-1932™) cells were a gift from Dr. Goutham Narla at the Case Comprehensive Cancer Center, Cleveland. VCaP and HCT-116 were gifted to Hera BioLabs in April 2017. MIA-PaCa-2, HCC1954, and 786-O were gifted to Hera BioLabs in June 2018. All cell lines were originally purchased from ATCC. VCaP and MIA-PaCa-2 cells were grown in Advanced DMEM (ThermoFisher #11995065) with 10% fetal bovine serum (Atlanta Biologicals # S12450) and 1% penicillin and streptomycin solutions (Cat# 15140–122, ThemoFisher). HCT-116 cells were grown in McCoy's 5a Medium Modified (ATCC #30–2007) supplemented with 10% fetal bovine serum (Atlanta Biologicals # S12450) and 1% penicillin and streptomycin solutions (Cat# 15140–122, Themofisher). HCC1954 and 786-O cells were grown in RPMI 1640 (ThermoFisher # A1049101)

with 10% fetal bovine serum (Atlanta Biologicals # S12450) and 1% penicillin and streptomycin solutions (Cat# 15140–122, ThemoFisher). All the cells were grown in a humidified incubator at 37°C with 5% $CO_2$. All cells lines underwent monthly testing for mycoplasma contamination (Lonza, LT07-710) and STR testing at later passages.

## Animal care and welfare

All animal studies were conducted under the authorities of University of Kentucky's and Icahn School of Medicine at Mount Sinai's IACUCs, who specifically reviewed and approved these protocols. Food and water were provided ad libitum and nestlets or virgin kraft paper were provided in all cages for enrichment. Seven SCID/NCr (CB17/Icr-*Prkdc*scid/IcrCr) male mice (BALB/c background, strain 01S11, The NCI Animal Production Program, Frederick, MD) were used for VCaP xenograft mouse study. All other mouse studies were performed in the NSG (NOD.Cg-*Prkdc*scid *Il2rg*tm1Wjl/SzJ, strain 005557 from The Jackson laboratory). Five NSG mice were used for HCT-116 xenograft development and 7 NSG mice were used for PDX establishment studies. Thirty four SRG rats were used for PDX establishment studies and thirty four SRG rats were engrafted with commercially available cell lines for xenograft development. Tumors were placed on the dorsal side of the flank so that they did not interfere with normal mobility or ability to nest properly. There were no changes observed in motility or food intake in tumor bearing animals. All animals were checked at least once daily for aspects of general health including activity, posture and fur grooming. Rats were also checked to ensure there is no porphyrin present. Body condition score was also assessed for mice and rats as previously described [17, 18]. Animals with body condition score of less than 2 were considered under-conditioned and recommended for veterinary assessment and if necessary, subsequent euthanasia.

Weight and tumor measurements (length and volume) were recorded 3 times weekly on Monday, Wednesday, and Friday. All animals were monitored once daily, including weekends and holidays, for general health, activity level, body and tumor appearance, mobility, and ability to eat, drink, and groom within normal limits. Tumors were monitored once daily for signs of ulceration. These measurements and all health observations were performed by trained animal care technicians and referred to veterinary staff if abnormalities were observed. Animal technicians are trained through AALAS courses, including "Post-Procedural Care of Mice and Rats in Research; Minimimzing Pain and Distress". In addition, animal technicians complete one-on-one hands-on training workshops to become adept at tumor measurements and monitoring, as well as monitoring clinical signs during daily health checks. The chart below indicates specific parameters that are monitored (Table 1).

Humane endpoints take the following into consideration: tumor diameter or tumor weight vs. body weight, tumor ulceration, animal weight, body condition score, animal mobility and activity. Animals are euthanized if tumors grow to longer than 40 in diameter on the longest edge, or when they reach 10% of the body weight (e.g. for a 200 g animal, the tumor cannot weigh more than 20 g = 20,000 mm³). Other criteria warranting euthanasia are ulceration on greater than 25% of the surface or perforation in any size ulceration that causes exudate or infection. If an animal loses greater than 20% initial body weight, they are euthanized. Euthanasia may also be carried out if the above are not met but the animal has reduced body condition score, hunched posture, ungroomed fur, porphyrin, respiratory abnormalities, impaired mobility or ability to perform daily tasks (eating, sleeping, ambulating) due to the position or size of the tumor, or unalleviated pain as suggested by the grimace scale and vocalization upon handling. If ulceration occurs, antibiotic ointment, which may include an analgesic, is applied topically at the onset of ulceration to prevent infection after consultation and approval by

**Table 1.**

| Parameter | Frequency | Scoring |
|---|---|---|
| Tumor length, width Volume = (L x W$^2$)/2 | Three times weekly | Absolute measurement |
| Body Weight | Three times weekly | Absolute measurement |
| Body Condition | Daily | 1–5 (Normal = 3) [17, 18] |
| Porphyrin | Daily | Presence (mild, moderate, severe) or absence |
| Activity level | Daily | Normal or reduced (mild, moderate, severe) |
| Posture | Daily | Normal or hunched |
| Fur/coat | Daily | Color; groomed or rough/ungroomed |
| Mobility | Daily | Normal vs. inhibited (describe how inhibited) |
| Tumor appearance | Daily | Normal or discolored or ulcerated. If ulcerated, estimate % surface ulcerated, indicate whether there is exudate or suspected infection |

veterinarians. Analgesics (meloxicam or carprofen) are administered as needed to manage suspected pain. If analgesics do not relieve pain as measured by clinical criteria mentioned above, the animal is euthanized. Animals were euthanized by $CO_2$, with secondary methods including cervical dislocation (all mice and rats <200g) or thoracotomy (rats >200g) as approved by the IACUC and in accordance with current AVMA guidelines.

## Tumor xenografts

For transplantation, 5-10x10$^6$ VCaP cells, 2x10$^6$ HCT-116 cells, 5 x10$^6$ MIA-PaCa-2, 10 x10$^6$ 786-O cells, or 5x10$^6$ HCC1954 cells were injected in each animal. Cells were resuspended in 250 μL of each cell line's respective culture media as listed in the Cell Culture methods. Immediately prior to injection, 250 μL 10 mg/mL Cultrex BME3 (Trevigen #3632-001-02) or Matrigel (Corning #354234) was added to the cell suspension for a final Cultrex or Matrigel concentration of 5 mg/mL. The suspension of cells and Cultrex/Matrigel was injected subcutaneously into the hindflank. Tumor growth was monitored by externally measuring the greatest longtitudinal diameter (length) and the greatest transverse diameter (width) using digital calipers (Fowler #54-100-067-1) 3 times a week. These measurements were used to calculate tumor volume by the modified ellipsoidal formula [19, 20]: *Tumor volume = (L x W$^2$)/2*. Studies were conducted after Institutional Animal Care and Use Committee approval and in strict compliance with institutional regulatory standards and guidelines.

## PSA analysis

Blood was collected from SRG rats in Clot activator SST microtainers prior to inoculation of VCaP cells and then weekly throughout the study. Blood was allowed to clot at room temperature for at least 30 minutes, then centrifuged at 6000xg for 3 minutes at room temperature to separate serum. Serum was analyzed for PSA by ELISA (ALPCO #25-PSAHU-E01) according to manufacturer's instructions.

## PDX implantation

Surgically resected NSCLC tissue from nine patients were obtained from the Biospecimen Procurement and Translational Pathology Shared Resource Facility of the University of Kentucky Markey Cancer Center (P30CA177558) in collaboration with Dr. Mark Evers under an approved University of Kentucky IRB application. De-identified patient samples were

provided to Hera Biolabs in DMEM + 10% FBS, 1% Penicillin/Streptomycin, 1% Amphoteri-cin B. These primary patient tumor tissues, annotated as passage 1 or P1, were sectioned into 2mm x 2mm pieces and implanted subcutaneously on the flank of SRG rats or NSG mice using a 10G trocar. Animals were treated with an analgesic (carprofen 5 mg/kg subcutane-ously) immediately post-surgery. Tumor diameter was measured using digital calipers 3 times a week. Tumor volume was calculated as $(L \times W^2)/2$, where width and length were measured at the longest edges [19, 20]. Humane endpoints follow PHS and AAALAC guidelines such that tumor length does not exceed 20 mm for mouse or 40 mm for rats, and the tumor volume does not exceed 10% of the total body weight (e.g. for a 200 g rat, tumor volume cannot exceed 20,000 mm$^3$). Other aspects of health were evaluated, such as body weight (20% body weight loss at any time point warrants euthanasia), body condition score, tumor ulceration, and ensuring that the tumor does not interfere with locomotion or normal activities. For tissue expansion, PDX tumors were excised from animals aseptically, sectioned into 2mm x 2mm pieces, and implanted into SRG rats and into NSG mice, using the same method. Subsequent passages from mouse/rat to mouse/rat are annotated as PDX passage 2, 3 or P2, P3, etc, to establish a bank of tissues. For the original patient tumor and at each passage in the animals, a small piece was fixed in 10% neutral buffered formalin for histology and flash frozen in 2-methylbutane chilled on dry ice or liquid nitrogen for genomic analysis.

## PDX genomic analysis

Genetic analyses were performed for three of the PDX lines at multiple passages for each line in addition to the original patient sample. DNA extracted from flash frozen PDX tissue using a Qiagen kit (Qiagen #69504) was sent to the University of Michigan MMGL-Molecular Genet-ics core facility for genomic analysis.

## Histology

Primary patient NSCLC tissue and NSCLC-PDX tissue collected from mouse and rat were fixed in 10% neutral buffered formalin, processed, paraffin embedded, and sectioned. Tissues were stained with H&E, P40, and TTF1. VCaP tumors grown in SCID/NCr mice or SRG rats were collected, fixed in 10% neutral buffered formalin, processed, paraffin embedded, sec-tioned, and stained for AR (ab108341, abcam) or PSA (A056201-2, Dako). For all staining, tis-sue slides were incubated with primary antibody overnight at 4˚C. DAB substrate was applied followed by counterstaining with hematoxylin. Tissue were stained for H&E by IDEXX or Icahn School of Medicine at Mount Sinai Pathology Core Facility.

## Western blot

A549 (ATCC® CCL-185™), PC3 (ATCC® CRL-1435™), and LNCaP/AR were used as positive and negative controls. LNCaP/AR cells were a kind gift from Dr. Charles Sawyers (Memorial Sloan Kettering Cancer Center, New York, NY). Tumors were homogenized and cell protein was isolated with RIPA Lysis and Extraction Buffer (ThermoFisher Scientific). Isolated protein was quantified, normalized by the Bio-Rad assay (Bio-Rad), run on a 12% SDS-PAGE (Invitro-gen, Life Technologies), and transferred onto Nitrocellulose Membranes (Bio-Rad). The mem-brane was blocked with 5% Nonfat Milk (LabScientific) in Tris-Buffered Saline–Tween 20 buffer. Membranes were probed with GAPDH (sc-32233, Santa Cruz) and AR (ab74272, Abcam). Membranes were exposed to ECL (Roche) following the manufacturer's instructions. Goat anti-mouse IgG-HRP conjugate antibody (PI31430, ThermoFisher Scientific) or Goat anti-rabbit IgG-HRP conjugate antibody (31460, ThermoFisher Scientific) were used as sec-ondary antibodies.

## Statistics

Graphpad Prism 7 was used to perform all statistical analyses. Two tailed *t*-tests (for two group comparisons) was used for experiments. P<0.05 was considered statistically significant. Data is presented as mean ± standard deviation or mean ± standard error of the mean as noted in the figures. Correlation measurements were obtained by Pearson Correlation.

## Results

### SRG rat lacks mature B and T cells, and has significantly reduced NK cells

The SRG rat carries an eight base pair deletion in the Rag2 coding exon and a sixteen base pair deletion in the first coding exon of the Il2rg gene (S1 Fig). Splenocytes, thymocytes, and whole blood were collected from wild-type and SRG rats and analyzed by flow cytometry to characterize the immune cell populations. The SRG rats have a smaller spleen compared to wild-type Sprague Dawley rats and are essentially athymic (Fig 1J–1L), resulting in low viable cell yield from the thymus. The wild-type Sprague Dawley rat thymus was comprised mostly of CD4/CD8 double positive cells whereas in the SRG rat, the viable cells recovered were CD4-/CD8- (Fig 1A–1C). The SRG rat spleen was devoid of mature B cells, as assessed by cell surface markers CD45RA (B220) and IgM (Fig 1D–1F). The SRG rat spleen also has lower NK cells compared to the wild-type rat spleen (6.11%± 1.98 vs. 1.31% ± 1.31, respectively; Fig 1G–1I).

Analysis of whole blood demonstrated that while a wild-type rat had 37.4% CD4+, 36.6% CD8+, 3.5% CD4+/CD8+ in the circulating lymphocytes (Fig 2A and 2C–2E), the SRG rat had significantly reduced populations at 1.6% CD4+, 5.3% CD8+, 1.2% CD4+/CD8+ cells (Fig 2B and 2C–2E). Similar to the SRG rat spleen, the SRG rat circulating blood was devoid of mature B cells, assessed by cell surface markers CD45RA (B220) and IgM (Fig 2F–2H). Strikingly, circulating NK cells in the peripheral blood of the SRG rat are significantly reduced (0.5%) relative to wild-type levels (10.1%) (Fig 2I–2K). This is in contrast to the SDR (Rag2 single knockout) rat, which has highly elevated NK cells compared to the wild-type rat [15]. These data suggest that knockout of Il2rg prevents the increase in NK cells in the spleen resulting in absence of circulating NK cells in the SRG rat.

### SRG rats support the growth of multiple human cancer cell lines

To determine if the SRG rats supports the growth of human xenografts, we inoculated several human cancer cell lines known to grow in immunodeficient mouse models. We examined the growth of the human colorectal carcinoma cell line, HCT-116, in SRG rats in comparison to NSG mice. We injected 2 x10$^6$ HCT-116 cells in Cultrex, an extracellular matrix protein isolated from Engelbreth-Holm-Swarm mouse sarcoma similar to Matrigel, into NSG mice and SRG rats. By 10 days post-inoculation, tumor take rate was 100% in both NSG mice and SRG rats. Despite the equal number of cells inoculated into the two species, SRG rats displayed increased growth kinetics. The NSG mice had tumor volumes of 1000–2700 mm$^3$ by 30 days post-inoculation and SRG rats had tumor volumes of 1800 mm$^3$ to over 12,000 mm$^3$ by 24 days post-inoculation (Fig 3A). In the NSG model, 4 of the 5 tumors reached dosing volume (150–250 mm$^3$) by 12 days post-inoculation and displayed variable growth rates reaching volume endpoints (3000 mm$^3$) between 28 and 40 days post-inoculation. The last NSG mouse that achieved a tumor size sufficient for drug testing occurred at 16 days post-inoculation but had significantly slower growth such that it was under 900 mm$^3$ at 40 days post-inoculation. In contrast, all 6 tumor HCT116 xenografts in SRG rats reached 150–250 mm$^3$ in volume by 10 days post-inoculation and exhibited faster growth kinetics compared with the mouse, with all tumors in the SRG rats reaching endpoints for size between 26 and 33 days after inoculation.

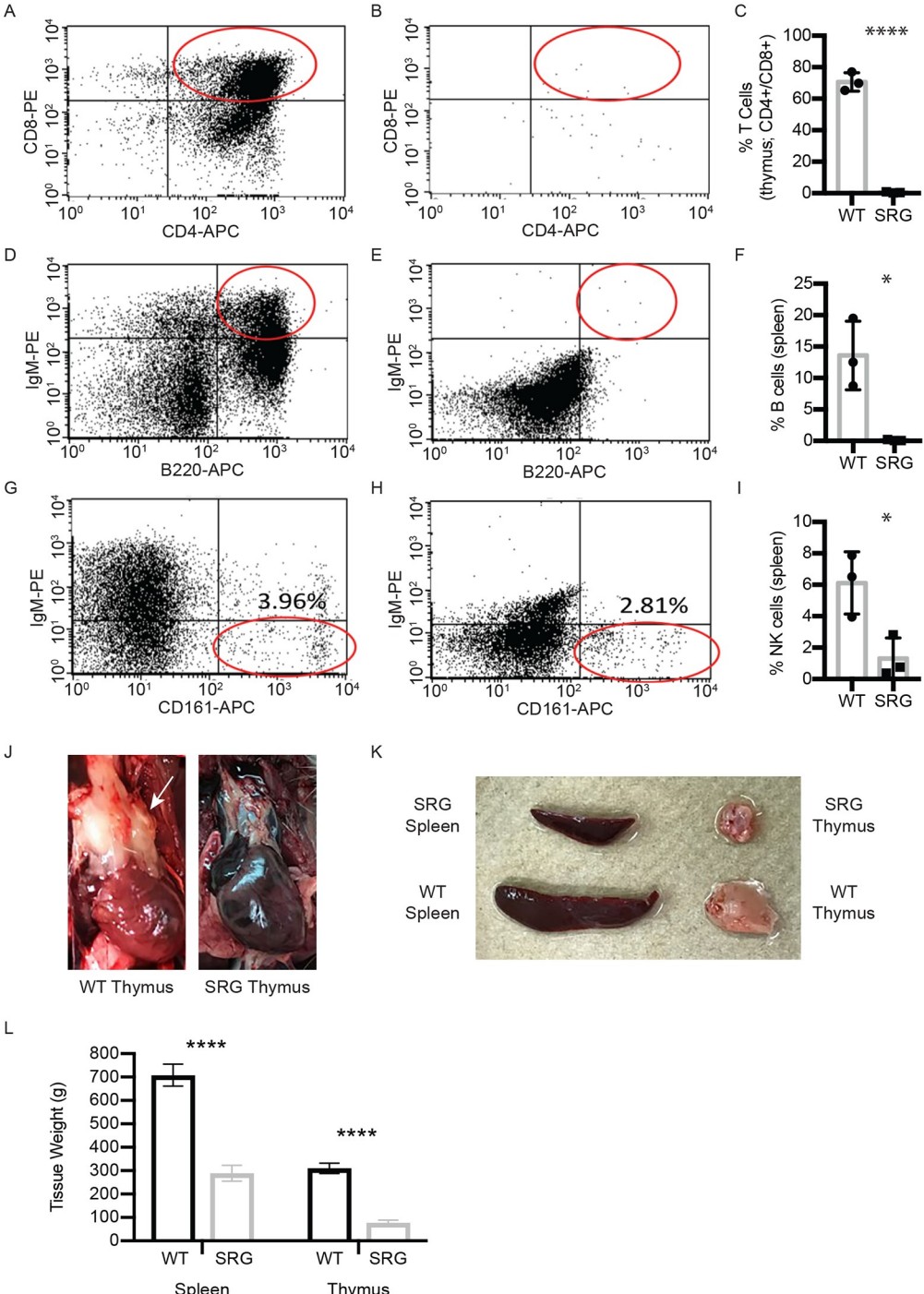

**Fig 1. Immunophenotyping of thymocytes and splenocytes in the SRG rat.** A-C) CD4+/CD8+ mature T cells in A) wild type control and B) SRG rat thymocytes. C) Quantification of data, n = 3, error ± SD. (Unpaired t-test, p-values: **** < 0.0001). CD4+/CD8+ mature T cells are absent from SRG thymocytes, compared to a wild-type control. The lack of thymus tissue in the SRG rat results in a low recovery of viable thymocytes. D-F) CD45R (B220)+/IgM+ cells in D) wild-type spleen and E) the SRG spleen. F) Quantification of data, n = 3, error ± SD. (Unpaired t-test, p-values: * < 0.05). Compared to B cells in a wild-type spleen, the SRG spleen contains no mature B cells as demonstrated by lack of CD45R (B220)+/IgM+ cells. G-I) NK cells in G) wild-type rat spleen and H) SRG rat spleen. I) Quantification of data, n = 3, error ± SD. (Unpaired t-test, p-values: * < 0.05). NK cells in the SRG rat spleen (H) are similar to or less than the amount of NK cells in the wild-type rat. The Il2rg knockout in the SRG rat results in significantly fewer NK cells than the single Rag2 knockout rat [8]. J) Image of wild-type Sprague Dawley versus SRG thymus. K) Images of wild-type

Sprague Dawley and SRG rat spleen. L) Quantitative comparison of wild-type Sprague Dawley versus SRG spleen and thymus at 8 weeks of age. Data represent average of 3 from each strain with SEM (Unpaired t-test, p-values: **** < 0.0001).

In addition to the colorectal cell line HCT-116, we also assessed the growth of several other cell lines in SRG rats (Fig 3B–3D): pancreatic cancer cell line MIA-PaCa-2, breast cancer cell line HCC1954, and renal cancer cell line 786-O. All cell lines tested engrafted with 100% efficiency in SRG rats and grew well over the study period. Growth kinetics of these human cancer cell lines have been tested by others in mouse xenograft models [21–24].

We next determined if SRG rats support the growth of human cancer cell lines that are difficult to grow in immunodeficient mouse strains. These include cell lines that have poor take rates or variable growth kinetics in mice. One such cell line is VCaP, a human prostate cancer

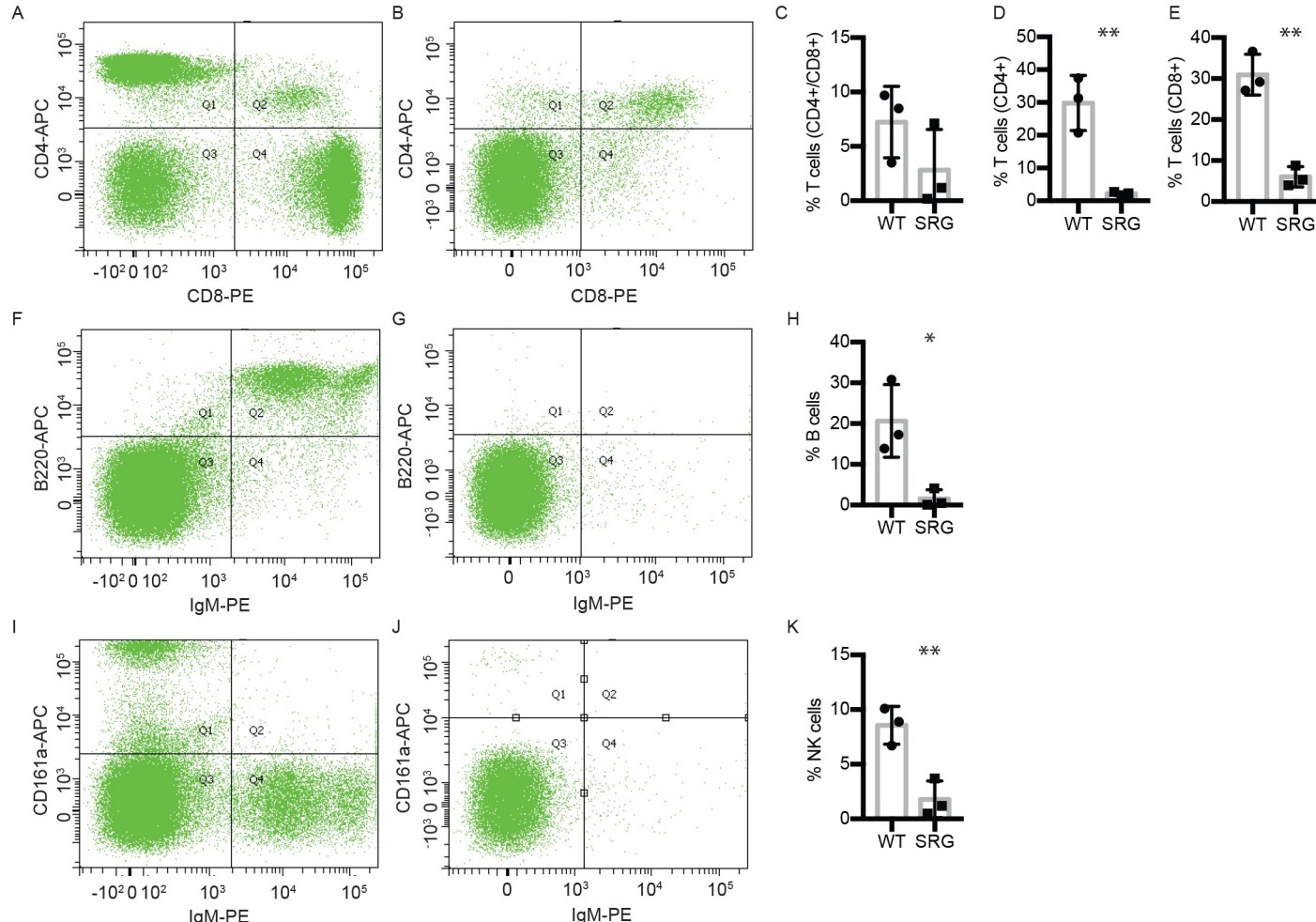

**Fig 2. Immunophenotyping of peripheral blood.** Flow cytometry dot plots show representative data from one WT and one SRG rat each. A-E) T cells in peripheral blood in A) wild-type rat and B) SRG rat. C-E) Quantification of data T cell populations, n = 3, error ±SD. (Unpaired t-test, p-values: ** < 0.01). T cells are significantly reduced in peripheral blood of the SRG rat (B; 1.6% CD4+, 5.3% CD8+, 1.2% CD4+/CD8+) compared to wild-type rat (A; 37.4% CD4+, 36.6% CD8+, 3.5% CD4+/CD8+). F-H) Circulating mature B cells in F) wild-type rat and G) SRG rat. The SRG rat is completely devoid of circulating mature B cells (G) compared to wild-type (F). H) Quantification of data, n = 3, error ± SD. (Unpaired t-test, p-values: * < 0.05). I-K) NK cells in the I) wild-type rat (10.1% CD161a+) and J) SRG rat (0.5% CD161a+). K) Quantification of data, n = 3, error ± SD. (Unpaired t-test, p-values: ** <0.01). Compared to NK cells in the wild-type rat (I; 10.1% CD161a+), the SRG rat has significantly reduced circulating NK cells (J; 0.5% CD161a+).

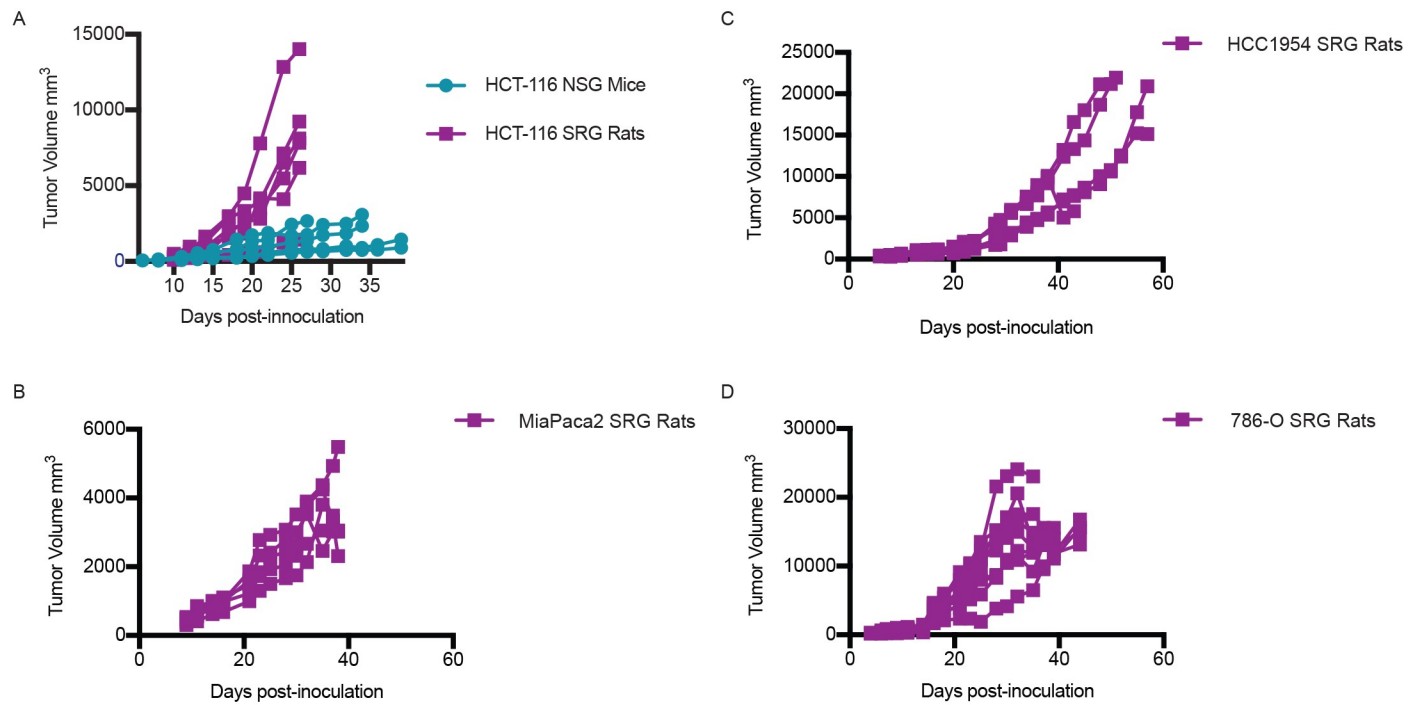

**Fig 3. Xenograft models in the SRG rat and NSG mouse.** A) Tumor growth curve in NSG mice and SRG rats inoculated with $2x10^6$ HCT-116 cells subcutaneously in the hind flank. Tumor width and length were measured three times weekly to calculate volume. B) Tumor growth curve in SRG rats inoculated with $5 x10^6$ MIA-PaCa-2 cells. C) Tumor growth curve in SRG rats inoculated with $5 x10^6$ HCC1954 cells. D) Tumor growth curve in SRG rats inoculated with $10 x10^6$ 786-O cells.

cell line derived from a vertebral metastatic growth. VCaP cells are difficult to maintain *in vitro* and display variable growth kinetics *in vivo* between published research groups [16]. We inoculated VCaP cells in Cultrex into the flanks of SCID/NCr mice, $5 x 10^6$ cells (~70,000 cells/cm$^2$ BSA), in keeping with published parameters. Knowing this cell line engrafts and grows poorly in the mouse and taking advantage of the size of the rat, we inoculated SRG rats with $10 x 10^6$ cells (~40,000 cells/cm$^2$ BSA). Pilot studies demonstrated that SRG rats had an 80% take rate. VCaP tumors surpassed 20,000mm$^3$ (humane endpoint) by 4–5 weeks post-inoculation in SRG rats, reaching adequate size for study evaluation between 17–23 days after inoculation (Fig 4A). In SCID/NCr mice, VCaP tumors engrafted in 60% of the mice but the growth kinetics were highly variable. The tumors did not reach dosing volume until 50 days post inoculation with a highly variable 30 day window for dosing enrollment (Fig 4A and S2 Fig). Overall, SRG rats demonstrated favorable take rate and growth kinetics for downstream efficacy studies. Individual level tumor measurements and body-weight data for all xenografts are provided in the supplemental tables.

Molecular analysis of VCaP tumors confirmed expression of the androgen receptor (AR) in both the SCID/NCr mouse and SRG rat models (Fig 4B and 4C). Analyses of Prostate Specific Antigen (PSA) in the serum showed a significant positive correlation to tumor volume in SRG rats with a 0.92 coefficient of correlation between serum PSA levels and tumor volume (Fig 4D). Serum PSA analysis was not performed for SCID/NCr mice since the procedure would require a terminal blood draw due to the much lower blood volume in the mice. Immunohistochemical analyses were used to further confirm the prostatic origin of these tumors. SCID/NCr mouse and SRG rat tumors expressed prostate-specific protein markers AR PSA (Fig 4E).

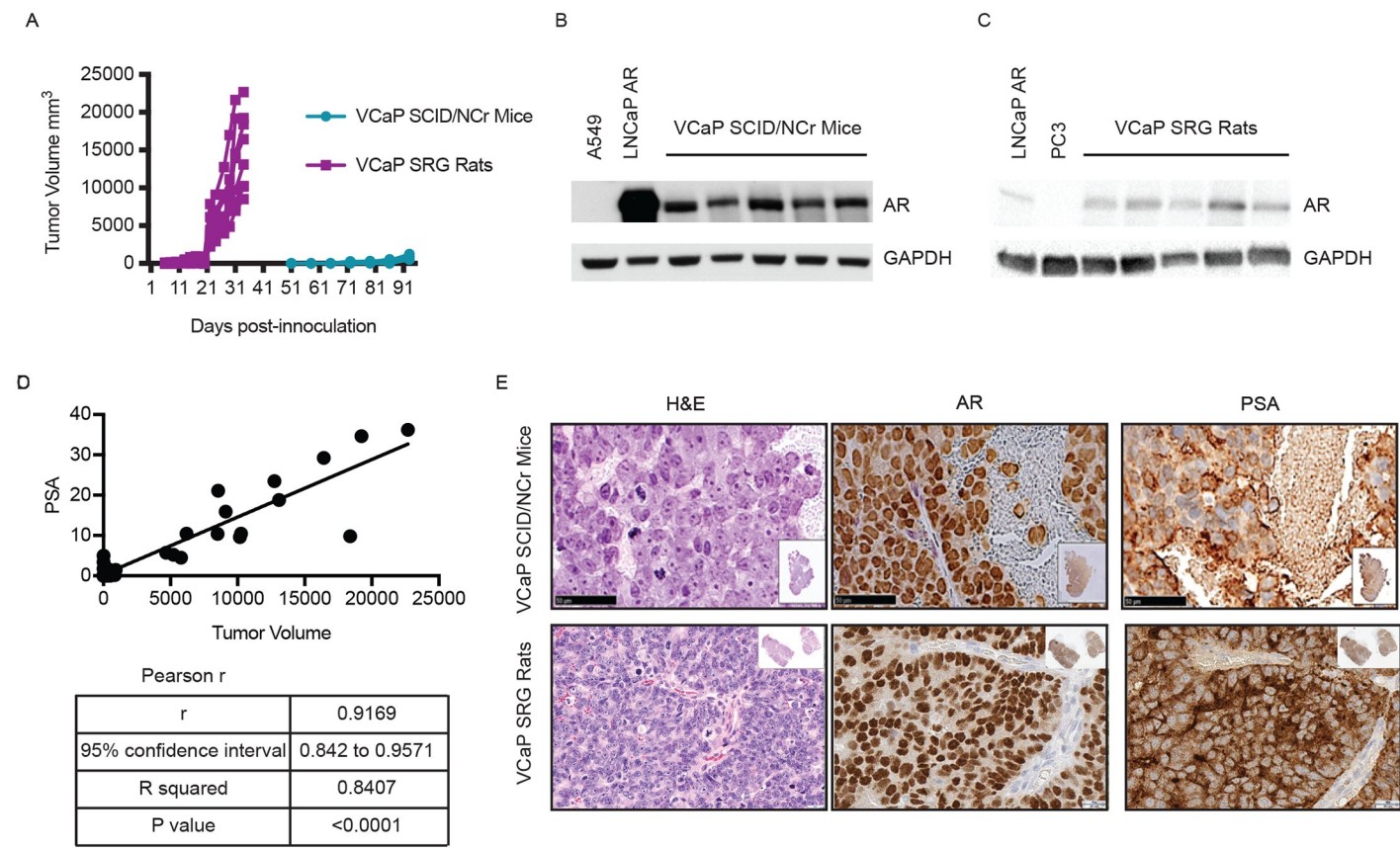

**Fig 4. VCaP xenograft model in SCID/NCr mouse and SRG rat.** SCID/NCr mice and SRG rats were inoculated with $5x10^6$ and $10x10^6$ VCaP cells, respectively, subcutaneously in the hind flank. Tumor width and length were measured three times weekly to calculate volume. A) Tumor kinetics in the SRG rat vs. SCID/NCr mouse. Each line represents tumor growth in an individual SRG rat or SCID/NCr mouse. B) Western blotting for AR in tumor tissue from the SCID/NCr mice. C) Western blotting for AR in tumor tissue from the SRG rat. D) Compilation of PSA in the serum of SRG rat inoculated with VCaP cells correlates with tumor volume. E) H&E staining and IHC staining for AR and PSA in VCaP tumor tissue from SCID/NCr mice and SRG rat.

These data demonstrate that SRG rats support the growth of human cancer cell lines with favorable take rates and growth kinetics for preclinical efficacy studies.

## SRG rats are permissive to PDX establishment and expansion

To determine if SRG rats support the growth of PDX, we transplanted primary tumors derived from patients with lung adenocarcinoma. For these initial studies, we implanted NSCLC samples surgically resected from nine different patients, of which seven successfully engrafted and grew to establish tissue banks for an overall 78% PDX establishment rate. Initial engraftment of the patient tumors resulted in tumor volumes of 4,000 mm³ by 75 days post-inoculation and tumor growth rate in SRG rats increased through serial passages (Fig 5A and S3A–S3F Fig). Immunohistochemical analyses of the tumors from the original patient sample and subsequent passages in SRG rats revealed comparable expression of p40 and thyroid transcription factor 1 (TTF1) confirming that the tumor maintained its histology *in vivo* (Fig 5B). Data in Fig 5 are from a single NSCLC patient tissue sample that was established as a PDX model in SRG rats (referred to as PDX 3010), however the data demonstrating congruency in immunohistochemistry are representative of the PDX lines we have established.

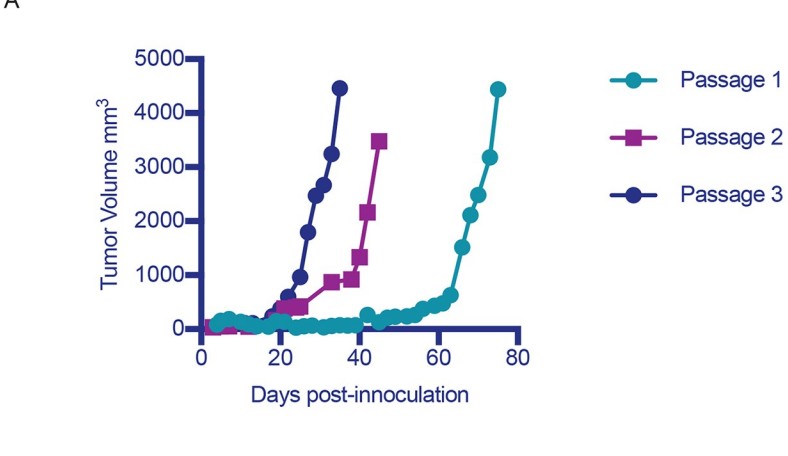

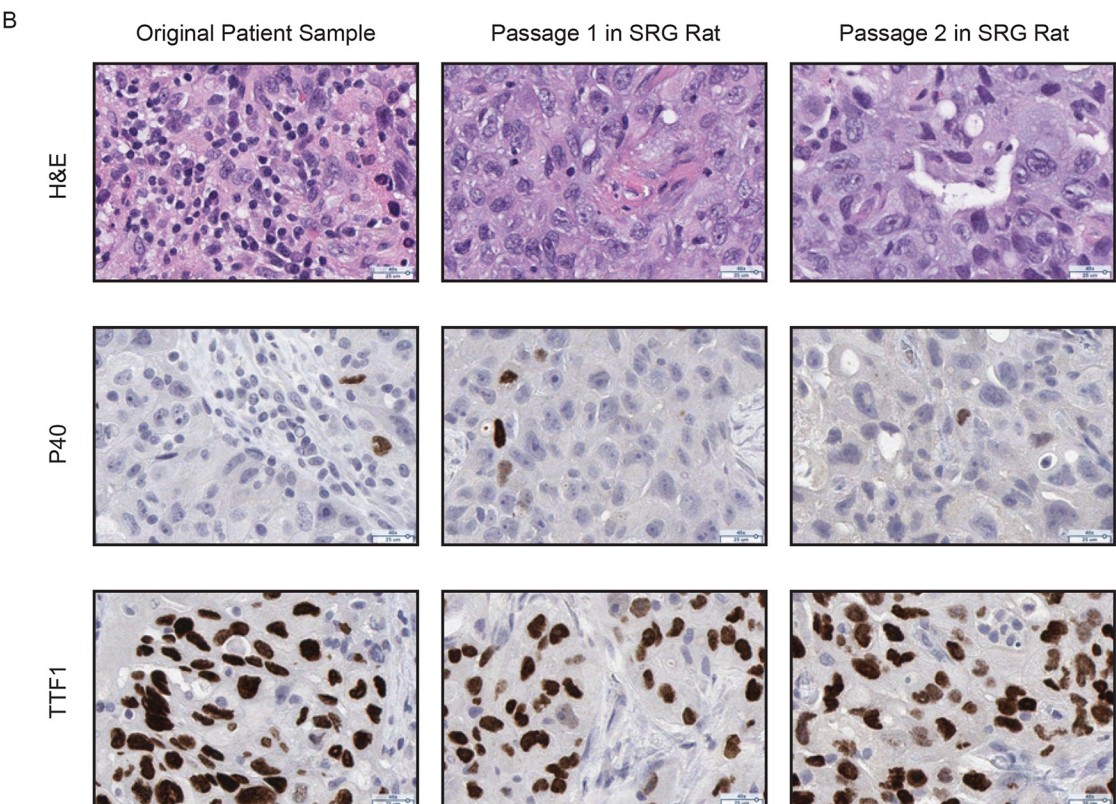

**Fig 5. PDX model in the SRG rat.** Patient derived lung tumor was implanted into SRG rats. A) Tumor growth curve shows multiple passages of the patient derived lung tumor from one patient and subsequent passages in the SRG rat. P1 is the initial passage in vivo in SRG rats. B) IHC staining for H&E, P40, and TTF1 in original patient tumor sample (3010), passage 1 of the same sample in SRG rat, and passage 2 of the same sample in SRG rat.

In order to evaluate the genomic instability of the PDX model, we performed next-generation sequencing after passages 1, 2 and 3 in SRG rats using a panel of 6000 genes, which are associated with a variety of cancers, for three of the seven PDX lines created (Table 2). Our analysis evaluated the variants detected within NSCLC-related genes such as KRAS, RAS, EGFR, and MET. A total of 87 SNPs were detected in 12 analyzed genes, of which 67 were conserved between original patient sample and subsequent passages (P1, P2, P3) in SRG rats for

**Table 2. Next generation sequencing results for PDX passages.**

| SNPs from P0-P3 | | | | |
|---|---|---|---|---|
| Gene | Conserved | Non-Conserved | Total SNPs | % Conserved |
| EGFR | 10 | 2 | 12 | 83.33 |
| KRAS | 5 | 0 | 5 | 100.0 |
| FGFR1 | 9 | 4 | 13 | 69.23 |
| PIK3CA | 9 | 2 | 11 | 81.82 |
| PTEN | 2 | 0 | 2 | 100.0 |
| ALK | 1 | 7 | 8 | 12.50 |
| EML4 | 0 | 1 | 1 | 0.0 |
| ERBB2 | 0 | 2 | 2 | 0.0 |
| AKT1 | 11 | 1 | 12 | 91.67 |
| MET | 1 | 0 | 1 | 100.0 |
| PPP2R1A | 17 | 1 | 18 | 94.44 |
| PPP2R1B | 2 | 0 | 2 | 100.0 |

77% concordance. All of the pathogenic SNPs identified in our analysis were conserved across all passages.

## Discussion

In this report, we characterized a Rag2, Il2rg double knockout rat model on the Sprague-Dawley strain, the SRG™ rat, and demonstrated that it is a competent host for human cancer cell lines, PDX modeling, and drug efficacy studies in oncology (SRG *OncoRat*®). We further demonstrated that SRG rats have high engraftment rates, favorable growth kinetics for efficacy studies, and can support large tumor volumes, providing ample tissue for molecular characterization and PDX bank establishment. Taken together, SRG rats are a valuable addition to the existing mouse models for use in preclinical oncology research. Furthermore, it has potential to function as a patient avatar whereby personalized genomically-guided precision therapies can be tested within a reasonable timeline to affect patient treatment.

Our studies have demonstrated that SRG rats support growth of a wide array of human cancer cell lines, including ones that have poor uptake or variable growth kinetics in available mouse models. Furthermore, the rat can humanely support the growth of tumors that are 10 times the size of those in the mouse, which allows for more tissue at study completion for downstream analyses. In addition, we can perform serial blood collection from the rat within humane guidelines to allow for pharmacokinetic and biomarker analysis following a single animal's progression throughout the study. The caveat to using the rat for efficacy studies is the need for larger amounts of dosing materials. However, our data suggest that for certain cell lines, fewer rats are needed to achieve a treatment cohort of tumor-bearing animals and the animal numbers can be further reduced compared to mouse due to the ability to perform serial blood draws in the same rat throughout a single study. For many human cancer cell lines, SRG rats display 80–100% engraftment and low variability in tumor growth leading to a high rate of enrollment into treatment studies allowing for a more defined and shorter enrollment window with fewer animals inoculated. Also, some compounds display low systemic exposures in mice making it difficult to perform efficacy studies; for these compounds an alternative model is available using SRG rats. Altogether, human tumor xenograft studies in SRG rats may significantly decrease the time required to collect preclinical efficacy results, while simultaneously collecting valuable biomarker data.

Our data also demonstrate SRG rats support PDX engraftment and may accelerate the timeframe for PDX propagation and preclinical efficacy studies. Published success rates for NSCLC-PDX establishment subcutaneously in mouse models are in the range of 20–40% which means even at best, the models fail for more than half of the patients [8–10]. We have demonstrated here that SRG rats can be used to establish NSCLC-PDX with 78% success as we were able to establish PDX banks from seven different patients of the nine patients sampled. Efficient take rates combined with the 10 times larger tumor size make SRG rats a promising tool for establishing PDX lines for a variety of populations and/or cancer-types. In addition, SRG rats can also greatly reduce the time needed to establish a PDX bank and can produce sufficient tumor tissue in the first *in vivo* generation (P1) to make a full preclinical study possible in the subsequent passage (P2). In comparison, mouse models may require multiple passages past the third or fourth before sufficient tumor-bearing animal numbers are produced for an efficacy study, increasing the risk of genetic drift from the original tumor. Since recent studies have shown that with each successive passage *in vivo*, tumors become more divergent genetically from the parent tumor, shortening the number of passages required to conduct an efficacy study will reduce animal numbers, associated costs, and ensure the tumors are more closely related to the parent tumor to better predict drug outcomes [11]. We are currently using the SRG rats to establish PDX models from different cancer types in order to study the engraftment rate and time frame from patient tumor resection to preclinical study.

In summary, we have created the immunodeficient SRG rat, a **S**prague-Dawley **R**ag2/Il2r**g** double knockout that lacks mature B and T cells and circulating NK cells. This model has been tested and validated for use in oncology (SRG *OncoRat*®) with different human cancer cell lines and PDXs. Our data demonstrate that the SRG rat has the potential to be a valuable model for evaluating drug efficacy in a wide range of human cancers.

## Supporting information

**S1 Fig. The SRG rat contains an 8bp deletion early in its single coding exon rendering the protein out of frame.** The SRG rat also carries a 16bp deletion in the first exon of the Il2rg gene to knock out its function.
(TIF)

**S2 Fig. Growth curve of VCaP in SCID/NCr mice.** Mean weight in mg with SEM.
(TIF)

**S3 Fig. Growth curve of multiple patient derived lung tumors in SRG rats.** Each graph depicts tumor volumes for individual animals for 6 different NSCLC patient samples (A-F). P1 is the inital implant into animals using fresh patient tissue, P2 is the first serial passage from animal to animal, P3 is the second serial passage from animal to animal. Sample 3067 (C) was implanted into SRG rats for P1 and then serially implanted into NSG mice for P2 and P3.
(TIFF)

**S4 Fig.**
(PDF)

**S1 Data.**
(XLSX)

## Acknowledgments

Dr. Innis acknowledges support from the Morton S. and Henrietta K. Sellner Professorship in Human Genetics. Sudeh Izadmehr acknowledges support from the National Institues of Health Loan Repayment Program.

## Author Contributions

**Conceptualization:** Goutham Narla, Tseten Y. Jamling.

**Data curation:** Fallon K. Noto.

**Formal analysis:** Fallon K. Noto, Jaya Sangodkar, Bisoye Towobola Adedeji, Sam Moody, Christopher B. McClain, Ming Tong, Eric Ostertag, Jack Crawford, Xiaohua Gao, Lauren Hurst, Caitlin M. O'Connor, Sudeh Izadmehr, Rita Tohmé, Jyothsna Narla, Kristin LeSueur, Kajari Bhattacharya, Amit Rupani, Marwan K. Tayeh, Jeffrey W. Innis, Matthew D. Galsky, B. Mark Evers, Analisa DiFeo, Goutham Narla, Tseten Y. Jamling.

**Investigation:** Jaya Sangodkar, Bisoye Towobola Adedeji, Sam Moody, Christopher B. McClain, Ming Tong, Xiaohua Gao, Lauren Hurst, Caitlin M. O'Connor, Sudeh Izadmehr, Rita Tohmé, Jyothsna Narla, Kristin LeSueur, Kajari Bhattacharya, Amit Rupani, Marwan K. Tayeh, Jeffrey W. Innis, Tseten Y. Jamling.

**Methodology:** Goutham Narla, Tseten Y. Jamling.

**Writing – original draft:** Fallon K. Noto, Jaya Sangodkar, Bisoye Towobola Adedeji, Goutham Narla, Tseten Y. Jamling.

**Writing – review & editing:** Erika N. Hanson.

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
