## [Decision Letter · Decision Letter 0]

10 Aug 2020

PONE-D-20-08672

The SRG rat, a novel Sprague-Dawley Rag2/Il2rg double-knockout validated for human tumor oncology studies.

PLOS ONE

Dear Dr. Noto,

Thank you for submitting your manuscript to PLOS ONE. After careful consideration, we feel that it has merit but does not fully meet PLOS ONE’s publication criteria as it currently stands. Therefore, we invite you to submit a revised version of the manuscript that addresses the points raised during the review process by both reviewers, experts in the field.

We look forward to receiving your revised manuscript.

Kind regards,

Francesco Bertolini, MD, PhD

Academic Editor

PLOS ONE

Journal Requirements:

2. Please provide additional information about each of the cell lines used in this work, including any quality control testing procedures (authentication, characterisation, and mycoplasma testing). For more information, please see http://journals.plos.org/plosone/s/submission-guidelines#loc-cell-lines.

3. At this time, we request that you  please report additional details in your Methods section regarding animal care, as per our editorial guidelines:

(1) Please state the source and number of mice used in the study  

Thank you for your attention to these requests.

4. To comply with PLOS ONE submission guidelines, in your Methods section, please provide additional information regarding your statistical analyses. For more information on PLOS ONE's expectations for statistical reporting, please see https://journals.plos.org/plosone/s/submission-guidelines.#loc-statistical-reporting.

6. Thank you for including your competing interests statement; "I have read the journal's policy and the authors of this manuscript have the following competing interests: Dr. Goutham Narla is on the scientific advisory board for HERA Biolabs."

We note that one or more of the authors are employed by a commercial company: Hera BioLabs Inc and Poseida Therapeutics Inc.

7.  Please amend your authorship list in your manuscript file to include author Erika N. Amini

8. Please amend the manuscript submission data (via Edit Submission) to include author Erika N Hanson

Reviewers' comments:

Reviewer's Responses to Questions

**Comments to the Author**

1. Is the manuscript technically sound, and do the data support the conclusions?

Reviewer #1: Yes

Reviewer #2: Yes

2. Has the statistical analysis been performed appropriately and rigorously? 

Reviewer #1: Yes

Reviewer #2: No

3. Have the authors made all data underlying the findings in their manuscript fully available?

Reviewer #1: Yes

Reviewer #2: No

4. Is the manuscript presented in an intelligible fashion and written in standard English?

Reviewer #1: Yes

Reviewer #2: No

5. Review Comments to the Author

Reviewer #1: In this paper, Noto FK et al, developed the SRG OncoRat model, a SCID rat on the Sprague-Dawley background that harbors a double knockout for the Rag2 and Il2rgamma genes. Similar to NSG mice, SRG rats showed enhanced immunodeficiency, lacking B, T, and NK cells. Combining these genetic changes in this immunodeficient rat allow the use of fewer animals through enhanced engraftment rates with more uniform tumor kinetics, and improved tumor growth profiles for both tumor cell lines and PDXs.

The paper discusses different aspects impacting the engraftment of human tumors: the comparison with NSG mice, the growth kinetics of several human cancer cell lines, the molecular characterization and histological analyses for PDXs...

Experiments are well conducted, the manuscript is clear, well written and the conclusions are supported by the data.

This reviewer only has minor points to be addressed:

- Figure legends 1 and 2: delete "252" and "278" refer to B cells and circulating mature B cells, respectively.

- Figure 5: the authors only showed a tumor growth curve from a single NSCLC PDXs established in SRG rats; please, provide growth curves for others 6 PDXs. This figure could also go to supplementary.

- To evaluate the genomic instability of the PDX model, the authors performed NGS after P1, P2 and P3, but any result is showed. Please, provide a figure or a table supporting this section.

- The authors assessed the growth of several tumor cell lines in SRG rats, including HCT-116, MIA-PaCa-2, HCC1954, and 786-O, but only the colorectal carcinoma cell line HCT-116 was also injected in NSG mice as comparison. They stated "growth kinetics of these human cancer cell lines have been tested by others in mouse xenograft models [14]", but any of these cell lines (MIA-PaCa-2, HCC1954, and 786-O) are mentioned in ref. 14. The authors should provide others refs. to demonstrate the higher engraftment rate of these cell lines in SRG rat in comparison to NSG mice.

Reviewer #2: In “The SRG rat, a novel Sprague-Dawley Rag2/Il2rg double-knockout validated for human tumor oncology studies” (PONE-D-20-08672), Noto et al generated SRG rat model carrying mutations in Rag2 and Il2rg. SRG rats had severe immunodificient phenotypes, i.e., lack of mature T/B cells and reduced NK cells. Then the authors demonstrated the advantages of using SRG rats in both cell-line-derived xenograft (CDX) and patient-derived xenograft (PDX) models. In CDX models, the engraftment rate (>90%) of VCaP cells was much higher than that of NSG mice. In PDX models, the authors achieved 7 PDX out of 9 samples from patients with lung adenocarcinoma. In both CDX and PDX models, SRG rats supported higher tumor growth-rate and easier sample collection, which makes them an alternative oncology model to mice.

This work is very interesting and adds important knowledge of immunodeficient rat models in oncology studies. However, the quality of this manuscript needs to be substantially improved before it’s accepted by Plos One.

The major issues:

1. Although the authors used “novel” in the title, similar models have already been characterized previously. These models include FSG (Prkdc/Il2rg; Mashimo et al. Cell Rep. 2012.), SD-RG (Rag1/Rag2/Il2rg; He et al. FASEB J. 2019) and RRG (Rag1/Il2rg; Ménoret et al. Transplantation. 2018). I highly recommend the authors to change the title and acknowledge the previous research in the manuscript.

2. The mouse strain NSG (NOD/LtSz-Scid Il2rg−/−) does not equal Nod-Scid. However, the authors may make the readers confused in the figure legends “Figure 4. VCaP xenograft model in NSG mouse and SRG OncoRat. NOD-Scid mice and SRG rats were inoculated with 5x106 and 10x106 VCaP cells, respectively, subcutaneously in the hind flank.” and “Supplemental Figure 2: Growth curve of VCaP in NOD-Scid mice. Mean weight in mg with SEM. Molecular analysis of VCaP tumors confirmed expression of the androgen receptor (AR) in both the NSG (NOD-scid) mouse and SRG rat models (Figure 4B,C).” To avoid misleading, I hope the authors to clarify which mouse strain was used in their experiments.

3. The statistical analysis and presentation should be improved. In the first paragraph of Results section, for example, the authors wrote “The SRG rat spleen also has slightly lower NK cells compared to the wild-type rat spleen (2.81% vs. 3.96%, respectively; Figure 1G -I).” 2.81% or 3.96% is just one representative result of the triplicate experiments. The authors should use the “Mean±SD” as shown in Figure 1I to describe this difference. The authors claimed that “Not only did the tumors grow faster in SRG rats, their individual growth kinetics were more uniform, leading to consistent tumor volumes throughout their growth.” However, I could not find any statistical analysis to verify the “uniform” growth kinetics.

4. In the last part of Results section, the authors did NGS to evaluate the genomic instability of their PDX models. However, there are no detailed results (figures or tables) or access number of the sequencing data. So I recommend the authors to provide the missing details in both Methods and Results sections.

The minor issues:

1. The authors verified the deletions in Rag2 and Il2rg respectively by Sanger sequencing. Their results would be solidified if they could examine the expression of these two genes at mRNA and protein levels.

2. Unlike Fig3A, Fig3BCD did not have NSG data as control. Although the authors cited previous results as “Growth kinetics of these human cancer cell lines have been tested by others in mouse xenograft models [14].”, I don’t think they are good controls because the experimental settings varied between different labs.

3. I highly recommend the authors to revise the manuscript carefully to avoid typographical or grammatical errors.

6. PLOS authors have the option to publish the peer review history of their article (what does this mean?). If published, this will include your full peer review and any attached files.

Reviewer #1: No

Reviewer #2: No

---

## [Author Response · Author response to Decision Letter 0]

4 Sep 2020

Note that these responses to reviewers are also found in a submitted Word document.

Reviewer #1: In this paper, Noto FK et al, developed the SRG OncoRat model, a SCID rat on the Sprague-Dawley background that harbors a double knockout for the Rag2 and Il2rgamma genes. Similar to NSG mice, SRG rats showed enhanced immunodeficiency, lacking B, T, and NK cells. Combining these genetic changes in this immunodeficient rat allow the use of fewer animals through enhanced engraftment rates with more uniform tumor kinetics, and improved tumor growth profiles for both tumor cell lines and PDXs.

The paper discusses different aspects impacting the engraftment of human tumors: the comparison with NSG mice, the growth kinetics of several human cancer cell lines, the molecular characterization and histological analyses for PDXs...

Experiments are well conducted, the manuscript is clear, well written and the conclusions are supported by the data.

This reviewer only has minor points to be addressed:

1. Figure legends 1 and 2: delete "252" and "278" refer to B cells and circulating mature B cells, respectively.

Response: We have deleted “252” and “278” from the figure legends.

2. Figure 5: the authors only showed a tumor growth curve from a single NSCLC PDXs established in SRG rats; please, provide growth curves for others 6 PDXs. This figure could also go to supplementary.

Response: We appreciate the suggestion.We have now added the growth curves for the other 6 PDXs.

3. To evaluate the genomic instability of the PDX model, the authors performed NGS after P1, P2 and P3, but any result is showed. Please, provide a figure or a table supporting this section.

Response: We appreciate the suggestion and we have added the following table to support the section:

Table 1:

4. The authors assessed the growth of several tumor cell lines in SRG rats, including HCT-116, MIA-PaCa-2, HCC1954, and 786-O, but only the colorectal carcinoma cell line HCT-116 was also injected in NSG mice as comparison. They stated "growth kinetics of these human cancer cell lines have been tested by others in mouse xenograft models [14]", but any of these cell lines (MIA-PaCa-2, HCC1954, and 786-O) are mentioned in ref. 14. The authors should provide others refs. to demonstrate the higher engraftment rate of these cell lines in SRG rat in comparison to NSG mice.

Response: We appreciate the suggestion and we have added additional references.

Reviewer #2: In “The SRG rat, a novel Sprague-Dawley Rag2/Il2rg double-knockout validated for human tumor oncology studies” (PONE-D-20-08672), Noto et al generated SRG rat model carrying mutations in Rag2 and Il2rg. SRG rats had severe immunodificient phenotypes, i.e., lack of mature T/B cells and reduced NK cells. Then the authors demonstrated the advantages of using SRG rats in both cell-line-derived xenograft (CDX) and patient-derived xenograft (PDX) models. In CDX models, the engraftment rate (>90%) of VCaP cells was much higher than that of NSG mice. In PDX models, the authors achieved 7 PDX out of 9 samples from patients with lung adenocarcinoma. In both CDX and PDX models, SRG rats supported higher tumor growth-rate and easier sample collection, which makes them an alternative oncology model to mice.

This work is very interesting and adds important knowledge of immunodeficient rat models in oncology studies. However, the quality of this manuscript needs to be substantially improved before it’s accepted by Plos One.

The major issues:

1. Although the authors used “novel” in the title, similar models have already been characterized previously. These models include FSG (Prkdc/Il2rg; Mashimo et al. Cell Rep. 2012.), SD-RG (Rag1/Rag2/Il2rg; He et al. FASEB J. 2019) and RRG (Rag1/Il2rg; Ménoret et al. Transplantation. 2018). I highly recommend the authors to change the title and acknowledge the previous research in the manuscript.

Response: We appreciate the suggestion and we have removed the word “novel” from the title and we have included the references.

2. The mouse strain NSG (NOD/LtSz-Scid Il2rg−/−) does not equal Nod-Scid. However, the authors may make the readers confused in the figure legends “Figure 4. VCaP xenograft model in NSG mouse and SRG OncoRat. NOD-Scid mice and SRG rats were inoculated with 5x106 and 10x106 VCaP cells, respectively, subcutaneously in the hind flank.” and “Supplemental Figure 2: Growth curve of VCaP in NOD-Scid mice. Mean weight in mg with SEM. Molecular analysis of VCaP tumors confirmed expression of the androgen receptor (AR) in both the NSG (NOD-scid) mouse and SRG rat models (Figure 4B,C).” To avoid misleading, I hope the authors to clarify which mouse strain was used in their experiments.

Response: We appreciate the helpful comment. To avoid misleading readers, we have clarified the mouse strain used in our experiment. We used SCID/NCr (CB17/Icr-Prkdcscid/IcrCr) male mice (BALB/c background, strain 01S11, The NCI Animal Production Program, Frederick, MD) for our VCaP xenograft mouse study.

We have also clarified the strain within the manuscript and figure legends:

Figure 4. VCaP xenograft model in SCID/NCr mice and SRG OncoRats. SCID/NCr mice and SRG rats were inoculated with 5x106 and 10x106 VCaP cells, respectively, subcutaneously in the hind flank.

Supplemental Figure 2. Growth curve of VCaP tumors in SCID/NCr mice. Mean weight in mg with SEM.

Molecular analysis of VCaP xenograft tumors confirmed the expression of the androgen receptor (AR) in the SCID/NCr mouse and SRG rat models (Figure 4B,C).

3. The statistical analysis and presentation should be improved. In the first paragraph of Results section, for example, the authors wrote “The SRG rat spleen also has slightly lower NK cells compared to the wild-type rat spleen (2.81% vs. 3.96%, respectively; Figure 1G -I).” 2.81% or 3.96% is just one representative result of the triplicate experiments. The authors should use the “Mean±SD” as shown in Figure 1I to describe this difference.

Response: We appreciate the suggestion and we have now used the “Mean ± SD” to describe the difference.

4. The authors claimed that “Not only did the tumors grow faster in SRG rats, their individual growth kinetics were more uniform, leading to consistent tumor volumes throughout their growth.” However, I could not find any statistical analysis to verify the “uniform” growth kinetics.

Response: We appreciate the suggestion and have removed the statement with regard to uniformity of the tumors.

5. In the last part of Results section, the authors did NGS to evaluate the genomic instability of their PDX models. However, there are no detailed results (figures or tables) or access number of the sequencing data. So I recommend the authors to provide the missing details in both Methods and Results sections.

Response: Please refer to Reviewer 1 Question 3.

The minor issues:

1. The authors verified the deletions in Rag2 and Il2rg respectively by Sanger sequencing. Their results would be solidified if they could examine the expression of these two genes at mRNA and protein levels.

Response: At this time, we have not examined expression at the mRNA and protein levels but feel that the immunophenotyping data demonstrates functional disruption of the genes.

2. Unlike Fig3A, Fig3BCD did not have NSG data as control. Although the authors cited previous results as “Growth kinetics of these human cancer cell lines have been tested by others in mouse xenograft models [14].”, I don’t think they are good controls because the experimental settings varied between different labs.

Response: While we understand that experimental settings could be varied amongst the different labs, the purpose of our studies was to confirm the growth of different cell lines from different tumor types in the SRG rats.

3. I highly recommend the authors to revise the manuscript carefully to avoid typographical or grammatical errors.

Response: We appreciate the suggestion and we will thoroughly revise the manuscript to avoid typographical or grammatical errors.

---

## [Decision Letter · Decision Letter 1]

22 Sep 2020

The SRG rat, a Sprague-Dawley Rag2/Il2rg double-knockout validated for human tumor oncology studies.

PONE-D-20-08672R1

Dear Dr. Noto,

We’re pleased to inform you that your manuscript has been judged scientifically suitable for publication and will be formally accepted for publication once it meets all outstanding technical requirements.

Kind regards,

Francesco Bertolini, MD, PhD

Academic Editor

PLOS ONE

Additional Editor Comments (optional):

Reviewers' comments:

Reviewer's Responses to Questions

**Comments to the Author**

1. If the authors have adequately addressed your comments raised in a previous round of review and you feel that this manuscript is now acceptable for publication, you may indicate that here to bypass the “Comments to the Author” section, enter your conflict of interest statement in the “Confidential to Editor” section, and submit your "Accept" recommendation.

Reviewer #1: All comments have been addressed

Reviewer #2: All comments have been addressed

2. Is the manuscript technically sound, and do the data support the conclusions?

Reviewer #1: Yes

Reviewer #2: Yes

3. Has the statistical analysis been performed appropriately and rigorously? 

Reviewer #1: Yes

Reviewer #2: Yes

4. Have the authors made all data underlying the findings in their manuscript fully available?

Reviewer #1: Yes

Reviewer #2: Yes

5. Is the manuscript presented in an intelligible fashion and written in standard English?

Reviewer #1: Yes

Reviewer #2: Yes

6. Review Comments to the Author

Reviewer #1: The authors have addressed all of my comments and concerns through their response and edits, and the manuscript is definitely improved. I have no further suggestions or corrections.

Reviewer #2: I carefully read the revised manuscript “The SRG rat, a Sprague-Dawley Rag2/Il2rg double-knockout validated for human tumor oncology studies” (PONE-D-20-08672R1). Noto et al made substantial improvement in revising the manuscript. I was satisfied with their responses to my previous comments. However, I hope the authors to be aware of the issues below.

1. I noticed that the authors used SCID/NCr mice as control for VCaP PDX. SCID/NCr mice lack T and B cells, but still have NK cells. SRG rats lack not only T and B cells due to Rag2 KO but also NK cells due to Il2rg KO. So the high growth-rate of PDX in SRG rats may be alternatively explained by this difference. I recommend the authors to address this in the text.

2. To benefit the field in rat PDX models, I highly recommend the authors to deposit their NGS data to the public server such as NCBI.

7. PLOS authors have the option to publish the peer review history of their article (what does this mean?). If published, this will include your full peer review and any attached files.

Reviewer #1: No

Reviewer #2: No

---

## [Editor Report · Acceptance letter]

24 Sep 2020

PONE-D-20-08672R1

The SRG rat, a Sprague-Dawley Rag2/Il2rg double-knockout validated for human tumor oncology studies.

Dear Dr. Noto:

I'm pleased to inform you that your manuscript has been deemed suitable for publication in PLOS ONE. Congratulations! Your manuscript is now with our production department.

Kind regards,

on behalf of

Dr. Francesco Bertolini 

Academic Editor

PLOS ONE